# Estimating the prevalence and risk of COVID-19 among international travelers and evacuees of Wuhan through modeling and case reports

George Luo[1]*, Michael L. McHenry[2], John J. Letterio[1,3,4]

**1** Case Comprehensive Cancer Center, Case Western Reserve University, Cleveland, Ohio, United States of America, **2** Department of Population and Quantitative Health Sciences, Case Western Reserve University, Cleveland, Ohio, United States of America, **3** Department of Pediatrics, Case Western Reserve University School of Medicine, Cleveland, Ohio, United States of America, **4** The Angie Fowler Adolescent & Young Adult Cancer Institute, University Hospitals Rainbow Babies & Children's Hospital, Cleveland, Ohio, United States of America

* gxl263@case.edu

**Data Availability Statement:** All relevant data are within the manuscript and its Supporting Information files.

## Abstract

Coronavirus disease 2019 (COVID-19) started in Wuhan, China and has spread through other provinces and countries through infected travelers. On January 23rd, 2020, China issued a quarantine and travel ban on Wuhan because travelers from Wuhan were thought to account for the majority of exported COVID-19 cases to other countries. Additionally, countries evacuated their citizens from Wuhan after institution of the travel ban. Together, these two populations account for the vast majority of the "total cases with travel history to China" as designated by the World Health Organization (WHO). The current study aims to assess the prevalence and risk of COVID-19 among international travelers and evacuees of Wuhan. We first used case reports from Japan, Singapore, and Korea to investigate the date of flights of infected travelers. We then used airline traveler data and the number of infected exported cases to correlate the cases with the number of travelers for multiple countries. Our findings suggest that the risk of COVID-19 infection is highest among Wuhan travelers between January 19th and 22nd, 2020, with an approximate infection rate of up to 1.3% among international travelers. We also observed that evacuee infection rates varied heavily between countries and propose that the timing of the evacuation and COVID-19 testing of asymptomatic evacuees played significant roles in the infection rates among evacuees. These findings suggest COVID-19 cases and infectivity are much higher than previous estimates, including numbers from the WHO and the literature, and that some estimates of the infectivity of COVID-19 may need re-assessment.

**Funding:** The authors received no specific funding for this work.

**Competing interests:** The authors have declared that no competing interests exist.

## Introduction

In December 2019, a novel strain of coronavirus (disease name, COVID-19) was identified in a group of patients in Wuhan, Hubei Province, China [1]. The pathogen spreads through human-to-human transmission, and data suggests that the pandemic has now affected a vast number of the world's population [2, 3]. On January 23rd, 2020, a quarantine was imposed on travel in and out of Wuhan to prevent the spread of COVID-19. However, it is estimated that more than five million residents had already left the city before the lockdown [4]. These travelers likely contributed significantly to the number exported cases in other countries. Additionally, national governments with citizens in Wuhan evacuated their citizens after the travel ban, and these evacuations led to the movement of infected individuals out of Wuhan. These two sources (infected Chinese travelers and infected foreign nationals) made up the vast majority of exported cases of COVID-19 from China during the early phases of transmission, and are designated by the World Health Organization (WHO) as "total cases with travel history to China" [5].

While we know that infected exported cases from Wuhan likely fall into these two groups, the number of risky travelers leaving Wuhan prior to the travel ban is difficult to estimate because the exact point in time at which they were exposed to COVID-19 or at significant risk of infection is poorly defined. While we may not know when travelers were first exposed to or infected by the virus, we want to define a critical period in which international travelers from Wuhan were likely already infected by the virus. This period of time was determined to be from January 19th to January 22nd, 2020 from our review of case studies.

The objective of our study is to estimate the COVID-19 infection rate of international travelers and evacuated citizens from other countries through analysis of exported infected cases from Wuhan. Our study utilized exported COVID-19 cases after the Wuhan travel ban which gives us insight into both the risk of exporting COVID-19 before the travel ban and the effects after the travel ban (i.e. those who were evacuated from Wuhan by their respective national governments).

Our paper's roadmap is organized as follows. We first analyzed the critical period when most infected cases were being exported from Wuhan, utilizing case reports in Japan, Singapore, and Korea. We then estimated an average infected rate among international travelers during the critical period of most infected exportation. Lastly, we investigated the infection rate among evacuees from Wuhan by each country of national origin. We compiled estimates of the number of infected in Wuhan from the literature and observe that our estimate is higher than previous reports. Overall, we highlight the critical period of exportation of COVID-19 from Wuhan and estimate an approximate infected rate of international travelers during that period with comparison to other estimates from the literature. In our discussion, we acknowledge the limitations in our study but also suggest that governments and researchers should collaborate to generate better estimates and predictions about COVID-19 and future epidemics. This would allow countries to be better prepared for future epidemics and outbreaks.

## Methods

### Exported cases of COVID-19

To ascertain exported cases of COVID-19, we used the "total cases with travel history to China" designation from the WHO reports, which are COVID-19 cases with recent travel history to China likely related to COVID-19 infection [5]. For WHO reports before February 3rd, 2020, that did not have a confirmed travel history to China, we used case reports and news reports (**S1 Table**) to track which cases were due to local transmission and excluded those

cases from the total number of cases in order to distinguish those who had contracted COVID-19 due to exposure in Wuhan.

## Case reports from the Ministry of Health in Japan, Singapore, and Korea

We relied on reports from national health agencies, containing information on the travel history of each patient, including the date that each case returned from Wuhan to their country of origin, to investigate when exportation of infected travelers occurred [6–8]. Each infected case that left China prior to February 14th, 2020 was defined as an exported case. Using this information, we plotted the flight date of infected individuals to see which dates had multiple exported cases (**S2 Table in S1 File**). To figure out the **critical period**, we investigated which flight dates had the highest number of exported travelers, which if summed together, account for the majority of cases. This critical period represents the highest rate of COVID-19 infection among international travelers from Wuhan.

## Linear regression modeling of exported travelers with estimated risky international travelers

To model the number of exported infected travelers with the number of risky international travelers, we utilized exported infection numbers and historical traveler data from Wuhan. For the exported infected cases, we used the total cases with travel history to China from the WHO technical report dated February 14th, 2020 [5]. Then for **infected travelers**, we subtracted the number of evacuees from the total infected cases for each country. For estimated travelers from Wuhan, we used a dataset containing 2 weeks of historical flight data from Wuhan to the top 30 countries by travelers in February 2018 [9]. We divided the number of travelers by 14 days (i.e. 2 weeks) to estimate daily travel out of Wuhan to other countries. The estimated risky international traveler population for each country is the product of the critical period and the estimated daily travelers from Wuhan.

For our linear regression model, we used the following equation and solved for the line of best fit for all countries besides Thailand. The infected rate of risky travelers corresponded to the slope of the line of best fit.

$$Y = \beta(X) + \alpha + \varepsilon \qquad (Eq1)$$

Where Y, β, X, represent the number of exported cases, infected rate, and number of risky travelers out of Wuhan (which is the daily travelers * critical period) respectively. We set α, our y-intercept, to zero because there is assumed to be no infected traveler without a risky traveler, and ε represents the difference between expected and actual values, i.e. the error term. We only used countries with 2 or more exported cases of COVID-19 in our linear regression model (more than 90% of confirmed exported cases) [5]. We chose to exclude countries with less than 2 exported cases because they either have few risky passengers or may be underestimating the cases of COVID-19. Data for Taiwan and Hong Kong did not differentiate between exported cases and local transmission, so they were also excluded.

We used Microsoft Excel and ggplot in R (version 3.6.1) to plot the graphs and find the correlation between infected and risky passengers. R was used to calculate confidence interval for the line of best fit. We used R to generate diagnostic plots of the linear regression of the infected travelers. We used these plots to determine outliers and if the model matches the assumptions of linear regression.

We used Inkscape to make a figure of two model scenarios: realistic scenario and the model scenario of COVID-19 infection rates of travelers in Wuhan. The realistic scenario assumes exponential growth which would be expected in the case of an epidemic [10]. Our model

scenario assumes all exported cases are from the critical time period and a constant infected rate. This simplifies estimation of the infection rate without other variables, which we do not have available. Therefore, the realistic scenario has an increasing infected rate over a longer period of time while our model scenario is a constant infection rate over a shorter critical time period.

### Evacuee data

For evacuees from Wuhan of each country, we compiled a table (**S3 Table in S1 File**) from individual news sources that included reports prior to February 14[th], 2020 and plotted the graphs of infected COVID-19 evacuees against the total number of evacuees.

### Inferring Wuhan infected numbers

To infer the number of infected cases in Wuhan, we assumed that the infected rate of travelers was the same as for the rest of Wuhan city [11]. Therefore, we multiplied the estimated infected rate of the infected travelers by the number of residents in Wuhan to estimate the number of infected individuals in Wuhan.

## Results

To explore the critical period of the exportation of COVID-19, we plotted the overall confirmed number of exported cases by confirmation date (**Fig 1A**). At the time of the Wuhan travel ban (January 23[rd], 2020), there were still relatively few confirmed exported cases, but the number of cases increases rapidly until around February 1[st], after which more cases are evacuees and local transmission between family members of Chinese travelers.

Since the incubation and subsequent detection of the virus may take over 5 days [12], we looked at over 40 case reports of exported COVID-19 cases from Japan, Korea, and Singapore to figure out the critical period in which infected travelers were exported. **Fig 1B** shows the flight dates of infected cases exported into each country, highlighting the period from January 19[th] to 22[nd], 2020 having the highest number of exported cases, corresponding to the four days before the travel ban. Flights on and after the 23[rd] of January are either indirect flights out of Wuhan or from other parts of China. Both of these graphs suggest that the vast majority of exported cases came during this short 4-day period.

To investigate the relationship between the number of travelers and exported cases, we estimated the number of travelers over 4 days from Wuhan based on historical flight data to each country and then compared it to the number of exported infected travelers, which is the exported cases minus infected evacuees (**Table 1**). We found a line of best fit (**Eq 1**) that accounted for most countries with multiple exported cases except Thailand (**Fig 2**). This line of best fit suggests that for most countries, there is an estimated 1 exported case for every 83 Wuhan travelers during the critical period. This also translates to an estimated 1.3% (CI 95%: 1.0–1.5%) infected rate among travelers from Wuhan during the critical period. However, Thailand, which has the most travelers from Wuhan, is a significant outlier among countries with multiple exported cases (**S2 and S3 Figs in S1 File**). This may be explained by the fact that most travelers from Wuhan to Thailand were tourists [20], and this population may not remain in Thailand even if they were infected.

We also compiled information about infected cases among evacuees of Wuhan (**Fig 3**). However, we found this data to be less predictable and uniform. For example, Germany, Japan, and Singapore have infected rates above 1% of the evacuees while other countries such as Australia and India have zero infected cases among more than five hundred evacuees each (**S3 Table in S1 File**). This is likely related to comprehensive COVID-19 testing of

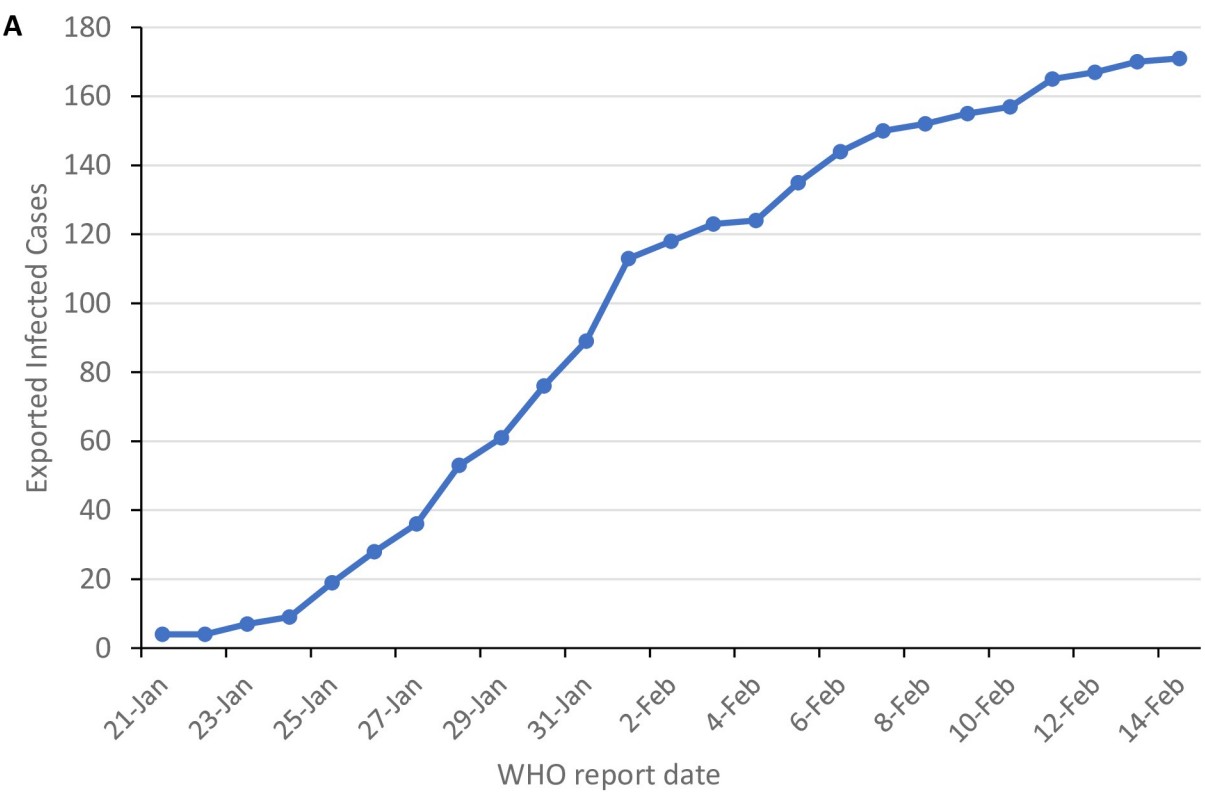

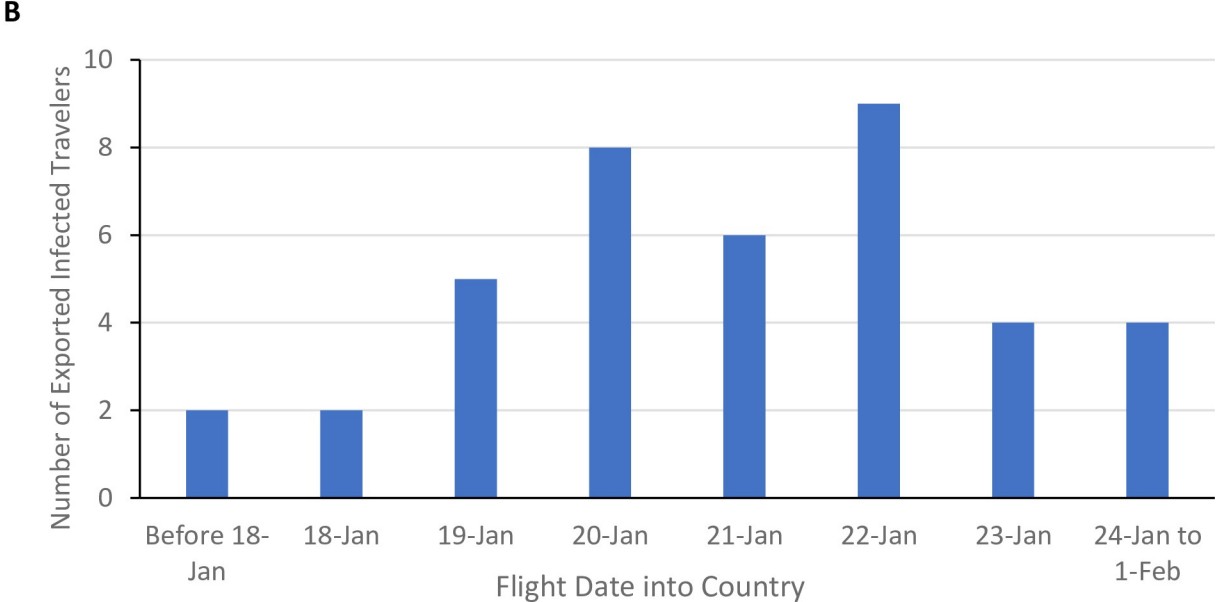

**Fig 1. Timing of exported infected cases. A.** Cumulative total of exported infected cases by confirmation date by WHO. **B.** Using case reports of individuals infected with coronavirus in Japan, Singapore, and Korea, a graph of the number of exported cases is plotted to the flight date into the country. Evacuees are excluded from this graph.

asymptomatic evacuees in Germany, Japan, and Singapore [21–23]. This could also be a result of a very heterogeneous population. While we cannot measure the impact of individual level risk factors, such as differences in hygiene practices, we examined whether other differences in

**Table 1. Exported infected cases from China.** The table shows number of exported cases from China to countries with multiple COVID-19 infected cases and differentiates them between infected evacuees and travelers. Estimated number of evacuees and estimated travelers during the critical period are also shown.

| Country | Exported COVID-19 Cases (Feb 14)[5] | Infected Evacuees (on Feb 14) ** | Total Evacuees ** | Infected Travelers (Exported Cases— Infected Evacuees) | Estimated Travelers from Wuhan for 4 days[10] |
|---|---|---|---|---|---|
| Singapore | 22 | 6 (5) | 266 | 17 | 739 |
| Japan | 24 | 9 | 763 | 15 | 1632 |
| Korea | 13 | 1 | 701 | 12 | 626 |
| Malaysia | 15 | 2 | 107 | 13 | 1155 |
| Australia | 15 | 0 | 500 | 15 | 1080 |
| Thailand | 23 | 0 | 138 | 23 | 4246 |
| Vietnam | 8 | 0 | 30 | 8 | 403 |
| India | 3 | 0 | 647 | 3 | 51 |
| Philippines | 3 | 0 | 30 | 3 | 232 |
| US | 13 | 3 | 800 | 10 | 695 |
| Canada | 6 | 0 | 398 | 6 | 226 |
| UAE | 6 | 0 | 0 | 6 | 417 |
| Germany | 2 | 2 | 124 | 0 | 99 |
| France | 5 | 0 | 302 | 5 | 265 |
| Italy | 3 | 1 | 56 | 2 | 203 |
| Russia | 2 | 0 | 140 | 2 | 84 |

** are data from **S3 Table in S1 File**.

the response to COVID-19 at the national level associated with different numbers of infected cases. Interestingly, many of the countries with earlier evacuations saw a greater number of infected cases while countries with later evacuations had fewer infected cases (**S3 Table in S1 File**). This holds true even after adjusting for the number of evacuees. Countries that did multiple evacuations such as Japan and Singapore saw a lower infected rate among the latter evacuations than the earlier ones.

To understand our estimated number of infections at the time of the Wuhan travel ban, we compared our estimates with others from the literature (**Table 2**). We observed a wide range of estimates from a few thousand infected cases in Wuhan to over a hundred thousand infected. This discrepancy includes many factors including methodology of estimate, timing of the data collection, and extrapolation of the estimates between reports but not a single factor fully accounts for the discrepancies. Our estimate extrapolated to the city of Wuhan would yield an approximate infected population of more than 140,000 cases at the time of the travel ban, among the highest of any published estimates that have been reported.

## Discussion

We acknowledge that there are a significant number of exported cases outside of the critical period suggested in our model but we believe that our model is still appropriate for estimating the number of infected cases during the period of highest risk. Additionally, we assumed a constant rate of infection among risky travelers during the critical period even though it is likely that the infection rate was increasing each day. Therefore, a realistic picture of the situation would be that there was a lower rate of infection before the critical period that contributed to the exported infections but that the infection rate at the time of travel ban approached our estimated rate of infection (**S1 Fig in S1 File**). Chinazzi's study on the transmission of COVID-19 also showed a similar critical period of five days of high COVID-19 exportation with a larger exported dataset and that the travel ban significantly decreased international

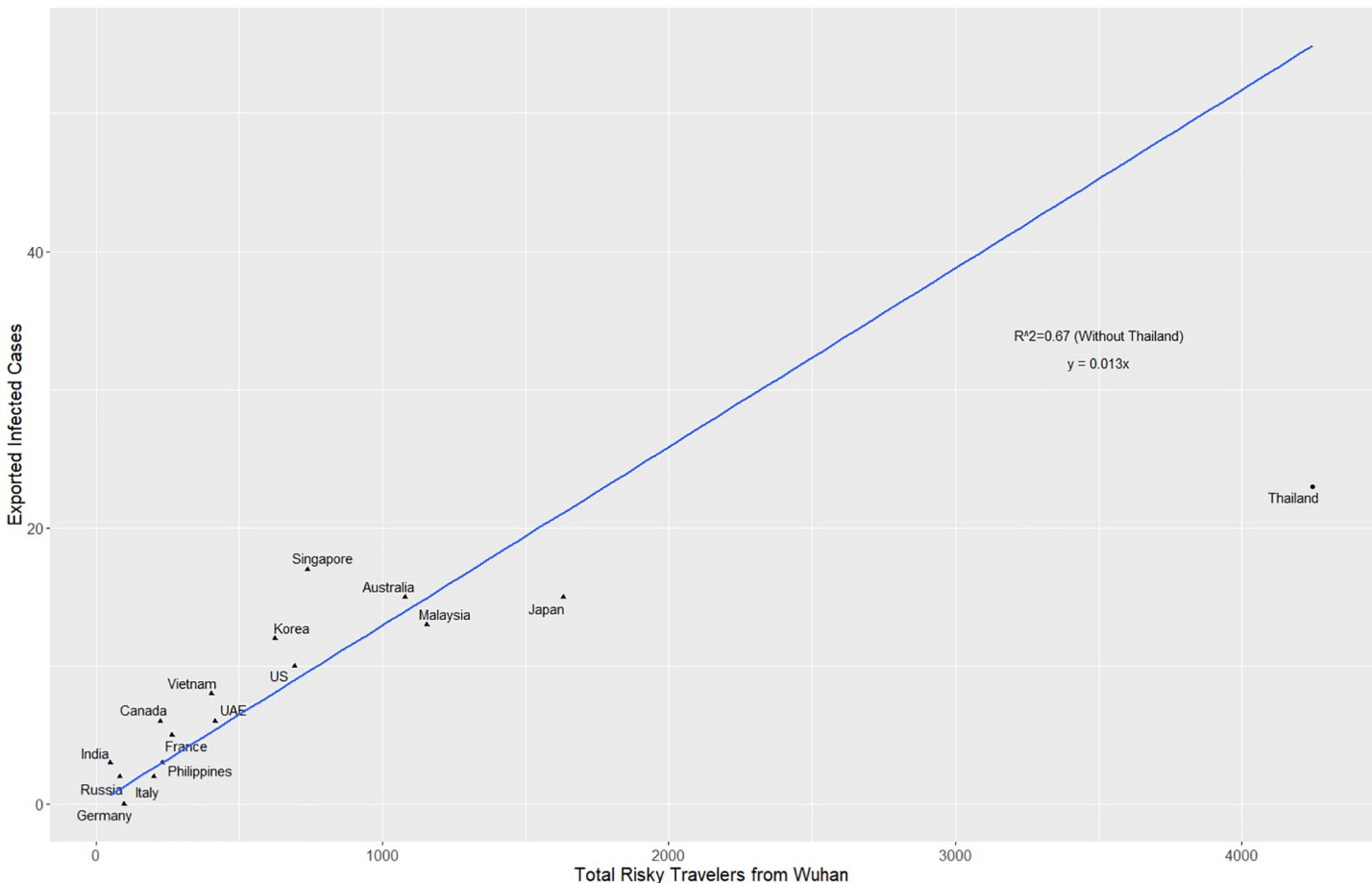

**Fig 2. Exported infected cases and risky travelers.** The number of confirmed exported cases were plotted against the number of estimated risky travelers from Wuhan from January 19th to 22nd, 2020. A line of best fit was used for countries besides Thailand. Infected evacuees were excluded from the exported infected cases.

transmission of COVID-19 [18]. We acknowledge that the critical period could be 4 or 5 days depending on the definition and modeling although the realistic scenario is likely that there was a continuously increasing infection rate among travelers before the travel ban.

Additionally, there is likely a discrepancy between the infection rates of travelers to individual countries. If Thailand is included in the linear modeling, the estimated infected rate of risky travelers drops to 0.8% which is much lower than the expected infected rate for the other countries. It is believed that this difference could be attributed to infected tourists traveling from Wuhan to Thailand who then returned to China before falling ill. Another potential explanation is that Thailand started temperature monitoring at their airports early in January 2020 and this may have reduced the transmission of COVID-19 among Chinese travelers [24]. Among other countries, there may be other factors affecting the number of infected exported travelers that are unaccounted-for, including COVID-19 testing methodology and criteria, length of traveler's stay, and underreporting of infection or infection symptoms. These factors may contribute to the heteroscedasticity observed in our model, but we still believe that besides Thailand, the variance of the residuals from the linear regression is reasonable for a model (**S3 Fig in S1 File**). We acknowledge our linear regression model has limitations such as omitted variable bias and unobserved heterogeneity, but despite the limitations, we believe our model

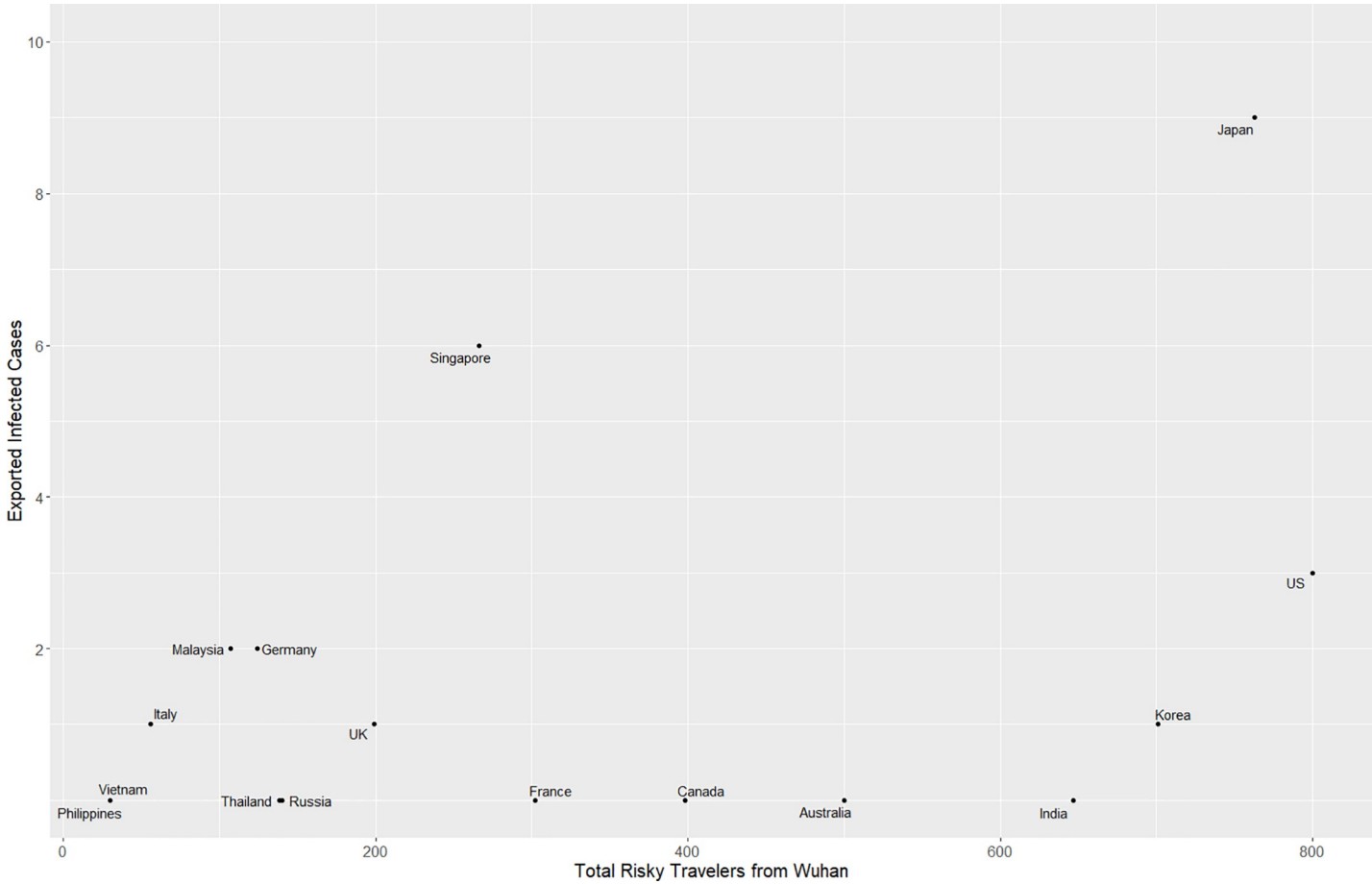

**Fig 3. Infected cases among evacuees of each country.**

is still better than the alternative to aggregate cases from various countries altogether as other papers have done [11, 13].

There are some assumptions in our model which have the potential to lead to an overestimate of the infected rate. There is likely transmission of COVID-19 between Chinese travelers

**Table 2. Published reports on estimating COVID-19 infection in Wuhan around the time of the travel ban.** Numerous reports have been published estimating the number of infected in Wuhan using a variety of methods. Growth modeling refers using epidemic growth modeling with an exponential growth phase. Exported cases utilized the number of exported cases from China to model the number of infected cases in Wuhan. SEIR refers to using a Susceptible-Exposed-Infectious-Removed' (SEIR) framework in the model. GLEAM stands for Global Epidemic and Mobility Model.

| Estimate around time of travel ban | Report | Estimation Method | First Publication Date |
|---|---|---|---|
| 4000 (for January 18th) | Imai et. al. 2020 (MRC Centre Reports) [13] | Exported Cases | January 22nd, 2020 |
| 21,022 | Read et al. 2020 (MedRxiv) [14] | Growth Modeling (SEIR) | January 28th, 2020 |
| 75,815 | Wu et. al. 2020 (The Lancet) [15] | Exported Cases | January 31st, 2020 |
| Model 1: 6924 | Jung et. al. 2020 (JCM) [11] | 1: Growth Modeling | February 14th, 2020 |
| Model 2: 19,289 | | 2: Exported Cases | |
| 21,675 | Wang et. al. 2020 (Cell Discovery) [16] | Growth Modeling (SEIR) | February 24th, 2020 |
| 16,589 | Lin et. al. 2020 (IJID) [17] | Growth Modeling (SEIR) | March 4th, 2020 |
| 117,584 | Chinazzi et. al. 2020 (Science) [18] | Growth Modeling (GLEAM) | March 6th, 2020 |
| 13,118 | Li et. al. 2020 (Science) [19] | Growth Modeling (SEIR) | May 1st, 2020 |

outside of China which would still be included in our exported cases of COVID-19. However, it might not be possible to tell whether a specific traveler was infected prior to or after leaving China, especially if they traveled with other infected cases. From case reports in Japan, Korea, and Singapore, we found that most exported cases were individual cases as opposed to large clusters, suggesting most infected cases likely became infected in China. We also used WHO data from February 14th, 2020 as a cutoff for exported cases because it is likely that any case after this date is not a direct export from Wuhan. Reassuringly, our estimated number of travelers to Thailand is within 5% of the actual numbers from Thailand's Ministry of Health reports [24].

The evacuee population is more difficult to model due to differences in procedures between their respective countries of origin. Each country has their own procedures for selecting when and how to evacuate people. Further, determination of infected status was not consistent across countries. Japan, Germany, and Singapore chose to test every evacuee, including asymptomatic people at the time of testing, which was not the case among other countries [21–23]. Additionally, each population of evacuees is unique and population-specific risk factors may lead to a different likelihood of getting infected for each population. However, the low number of infected cases from the later flights suggest that transmission of the virus is likely slowed by measures taken by the travel ban. Infected cases from before the travel ban are more likely to be symptomatic and would have been unable to evacuate at a later date, leading to the possibility that the proportion of symptomatic and asymptomatic carriers was systematically different among early and late evacuees. Our estimated infected rate for travelers is similar to the infected rates among evacuees from Germany, Japan, and Singapore, lending weight to the validity of our estimates.

Our estimated rate of infection among risky international travelers from Wuhan was 1.3% which is about 140,000 infected when extrapolated to the city of Wuhan. Notably, this is much higher than the other reported estimated numbers of infections in Wuhan around the time of quarantine [11, 13–18]. Our approach is unique in that we've utilized publicly available data about travelers and exportation of COVID-19 to each country and correlated that to an estimated infected rate. We believe our approach is more accurate than simply using all travelers and exported cases because it better adjusts for underreporting, inadequate testing, or other factors affecting accurate COVID-19 numbers by individual countries.

Our estimate is similar to estimates published in prior studies but still higher than those listed in our table. One key question is whether this infected rate can be extrapolated to the general population of Wuhan around the time of the travel ban. Other studies have used exported cases to infer a higher number of infected cases in the Wuhan area around the time of the travel ban than the confirmed numbers [11, 13, 15]. Although there will be some differences between travelers to Wuhan and the local population, some of which may confer differential risk profiles for each population, we believe this is the best estimate according to available data. If so, an estimated 110,000–160,000 out the 11 million people in Wuhan may have been infected at the time of the travel ban. This number would represent those infected by the virus that eventually develop COVID-19 symptoms but may not include asymptomatic patients, who can still transmit the virus [25]. COVID-19 infection data from the Diamond Princess Cruise ship suggests 18% of all COVID-19 cases may be asymptomatic [26], so this population may contribute to the underestimation of exported cases and unsuspected transmission of COVID-19. Our estimated COVID-19 infected numbers are higher than previous reports as we use data up to February 14th, 2020 to infer the infected rate around the time of the travel ban. We believe this is more accurate estimate because previous reports using exported infected cases were published using data before the full incubation period of

COVID-19, which has been demonstrated to be an average of 5.1 days with examples up to 2 weeks in some cases [2, 12].

The total confirmed infected cases worldwide in the WHO report only number around 70,000 as of February 17[th] [27]. Yet, our estimates suggest that true infected cases were potentially much higher. The travel ban from Wuhan seems to be effective in reducing exported cases and global spread would appear to have been much worse had the travel ban occurred any later. Additionally, from evacuee data, the infected rate in Wuhan appeared to remain high the week following the travel ban but potentially decreased afterwards. Our findings suggest that the COVID-19 infections in Wuhan may be much higher than previous estimates based on the exported cases from Wuhan around the time of the travel ban. This information and research can be used by governments and health organizations for better assessment of the true infected population, allowing government agencies, healthcare providers and organizations, and researchers better allocate resources and enact response measures during the COVID-19 pandemic and future epidemics.

We would recommend government and researchers to work together in the future to better assess the COVID-19 situation and any future epidemics. For instance, in the scenario of COVID-19, it was not easy to obtain information about exported cases of COVID-19 in each country about specific date of entry and date of symptoms. The WHO reports contain aggregate information about COVID-19 cases, but it is not always appropriate to aggregate the cases because each country's policy, healthcare, and demographics are different from one another. Additionally, there are examples of cases where infected travelers from Wuhan had COVID-19 but returned to China before being detected [28]. This type of information and accurate exported case data would be helpful in estimating the prevalence of the disease but is not easily obtainable to researchers. In the future, better access and aggregation of high-quality data will not only speed up research about the situation but also improve the quality of the research. Governments need to be supportive of this research for it to work by providing real-time, high-quality, and honest data. Ideally, research and predictions can be updated on a weekly basis using the latest information. This would require setting up a model that incorporates the information and is prepared for the future epidemics as well cooperation from government agencies from multiple countries. In our exported cases linear model, it requires air travel data, the number of exported cases by air travel, and a time frame of sufficient cases which is simple and can be updated easily for a quick estimation. However, researchers may need to adapt or develop alternative models depending on the data availability and circumstances to estimate infections. By having these models and estimates, governments will be better prepared to make policies such as quarantine and travel bans to handle the predicted situation and prevent exacerbation of a problem.

## Conclusions

In our study, we used exported COVID-19 cases from China to estimate the infected rate among international travelers from Wuhan, China. We used a linear regression model with countries having multiple exported cases to compare the number of exported cases with the number of travelers from Wuhan. Our analysis suggests that except for Thailand, an estimated 1.3% of international travelers from Wuhan were infected in the four days before the travel ban. Furthermore, evacuee data from Wuhan was heterogeneous but multiple countries, including those tested all evacuees for COVID-19, saw more than 1% of their evacuees infected. Extrapolating the international traveler data to the general Wuhan population suggest an estimated 140,000 people were infected at the time of the Wuhan travel ban which is higher than previous estimates. We suggest researchers to revisit estimates and models of the early

COVID-19 transmission for better understanding of the initial epidemic of COVID-19. Additionally, we suggest that government and researchers should collaborate in modeling to prepare for future epidemics and pandemics.

## Supporting information

**S1 Table.**
(XLSX)

**S1 File.**
(PDF)

## Author Contributions

**Conceptualization:** George Luo, John J. Letterio.

**Data curation:** George Luo.

**Formal analysis:** George Luo.

**Investigation:** George Luo.

**Methodology:** George Luo, Michael L. McHenry.

**Project administration:** George Luo.

**Resources:** George Luo, John J. Letterio.

**Software:** George Luo.

**Supervision:** John J. Letterio.

**Validation:** George Luo, Michael L. McHenry.

**Visualization:** George Luo.

**Writing – original draft:** George Luo.

**Writing – review & editing:** George Luo, Michael L. McHenry, John J. Letterio.

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
