## [Decision Letter · Decision Letter 0]

26 Mar 2020

PONE-D-20-05233

Estimating the prevalence and risk of COVID-19 among international travelers and evacuees of Wuhan through modeling and case reports

PLOS ONE

Dear Mr. Luo,

Thank you for submitting your manuscript to PLOS ONE. After careful consideration, we feel that it has merit but does not fully meet PLOS ONE’s publication criteria as it currently stands. Therefore, we invite you to submit a revised version of the manuscript that addresses the points raised during the review process.

I have now received the opinion of two subject matter experts. Please be advised that my decision (and the reviewer's recommendations) would have been different (tending to reject), had it been another topic. However, I do believe that your research is relevant and it will contribute to the better understanding of the current emergy situation throughout the world. The paper needs to be re-written for the sake of clarity, with careful attention to the methods section. If you need professional proofread services, please contact the journal office. Please follow the reviewer's recommendations. Let me highlight the most critical issues (where R2#3 is the third comment of reviewer 2). 1. Review of the literature: table with other papers examining similar issues. (R1#1, R2#1)2. Data sources table and output table R2#3 R2#83. Clarify methods and models. R2#5, R2#6, R2#124. Discussion: Please focus your discussion on your results and your limitations. One of them is the incubation period, which according to Bati et al (2020) the median incubation period was estimated to be 5.1 days.5. Future lines of research: Please suggest, based on your findings and limitations, what/how should researchers continue investigating. For example, R2#11 is a relevant issue for further research. ReferencesBai, Yan, et al. "Presumed asymptomatic carrier transmission of COVID-19." *Jama* (2020). DOI: 10.7326/M20-0504 

We would appreciate receiving your revised manuscript by May 10 2020 11:59PM. To enhance the reproducibility of your results, we recommend that if applicable you deposit your laboratory protocols in protocols.io, where a protocol can be assigned its own identifier (DOI) such that it can be cited independently in the future. For instructions see: http://journals.plos.org/plosone/s/submission-guidelines#loc-laboratory-protocols

We look forward to receiving your revised manuscript.

Kind regards,

Jordi Paniagua

Academic Editor

PLOS ONE

Journal Requirements:

Reviewers' comments:

Reviewer's Responses to Questions

**Comments to the Author**

1. Is the manuscript technically sound, and do the data support the conclusions?

Reviewer #1: Partly

Reviewer #2: Partly

2. Has the statistical analysis been performed appropriately and rigorously? 

Reviewer #1: Yes

Reviewer #2: No

3. Have the authors made all data underlying the findings in their manuscript fully available?

Reviewer #1: Yes

Reviewer #2: Yes

4. Is the manuscript presented in an intelligible fashion and written in standard English?

Reviewer #1: Yes

Reviewer #2: No

5. Review Comments to the Author

Reviewer #1: The authors used the number of exported COVID-19 cases from Wuhan to overseas and the number of infections/cases detected among passengers on charter flights by different countries to estimate the prevalence of COVID-19 in Wuhan. However, the same analysis has been done by several groups. They have either published their papers or uploaded their results in preprint archives. I suggest the authors do a thorough search of the similar studies and summarize them in a table to highlight the similarities and differences among different studies.

Reviewer #2: The aim of this paper is to analyse the critical period when the majority of Covid-19 infected cases were being exported, utilizing case reports in Japan, Singapore, and Korea. Moreover, it also estimates an average infected rate among international travelers during the risk period and investigates the infection rate among evacuees from Wuhan by each country. While the paper studies a current topic and it is very interesting, it would benefit from a major revision:

1) The introduction sets the topic and explains the outbreak of Covid-19 and the measures implemented in Wuhan. However, it does not say how it contributes to the previous literature and the importance of this study. Covid-19 is still a new virus but there are already lots of paper analysing it and there is previous literature in epidemics that can help you position your paper on the literature.

2) The objective of the paper needs to be clearly stated in the introduction.

3) The main variable is “total cases with travel history to China” – you need to define it properly in the text. Moreover, the data source on all the variables is needed. Provide a table with all the relevant information and cases.

4) The figures have a very low quality.

5) Equation 1 – this is not the best way to specify a model. You need to show the dependent variable and the covariates with their coefficients and the error term. Notation is important to follow the paper.

6) The methods are not clear. They are misleading and they don’t really match what you do in the results. It is important that you rewrite the methods section. The results are interesting, but they need to be well supported by the methods.

7) I would suggest that you say only once the software used (excel or R).

8) While the journal requires you to publish the dataset, I would suggest you create nice tables out of the two excels that you are uploading as supplementary material. They could be very informative. You need to show a descriptive table – number of cases, exported, travelers…etc.

10) It looks like from the data, that most of the infected cases traveled 4 days before the ban – however, you have a very small sample. Moreover, the incubation period is not known, so these people could have been infected long before traveling.

11) You analyse the exports from Wuhan, but since the outbreak and before the van, people in China were moving around. Flights from other regions in China to different countries could also carry infected people. Could that be checked too?

12) In the discussion, you talk about a model, but the model is not really explained in the methods section.

13) While the discussion is interesting, you need to write down the main limitations. You are looking at the correlation of two variables – so, you might have omitted variable bias. You don’t know the incubation period. And other data limitations that you faced during the study.

14) The paper would benefit from a full rewording – in many cases it feels like the authors are listing concepts (mainly in the methods and results). The text should flow.

6. PLOS authors have the option to publish the peer review history of their article (what does this mean?). If published, this will include your full peer review and any attached files.

Reviewer #1: No

Reviewer #2: No

---

## [Author Response · Author response to Decision Letter 0]

10 May 2020

Response to Revision Requests

Reviewer #1 Comments

I suggest the authors do a thorough search of the similar studies and summarize them in a table to highlight the similarities and differences among different studies.

Thank you for your comments. We have organized and summarized a table of published or preprinted estimates of the Wuhan infections in our revision (Table 2). We have highlighted our estimated number of infections is on the higher side of the estimates but not out of proportion. 

Reviewer #2 Comments

1) The introduction sets the topic and explains the outbreak of Covid-19 and the measures implemented in Wuhan. However, it does not say how it contributes to the previous literature and the importance of this study. Covid-19 is still a new virus but there are already lots of paper analyzing it and there is previous literature in epidemics that can help you position your paper on the literature.

We agree with your comments about positioning our paper in regard to other estimates. We discuss the value added of our manuscript in the revised introduction (lines 73-76). We have also included a summary of the literature to highlight how our manuscript differs from previous publications (Table 2). 

2) The objective of the paper needs to be clearly stated in the introduction.

We have added the objective of our paper to the introduction (lines 65-66). 

3) The main variable is “total cases with travel history to China” – you need to define it properly in the text. Moreover, the data source on all the variables is needed. Provide a table with all the relevant information and cases.

We have added definitions to the variables (lines 80-82, 91-94, 99-100). We have added Table 1 with all the information used to graph figure 2 and 3. Supplementary Table 1 has a list of data complied used to graph Figure 1A. Supplementary Table 2 has a list of data and sources to plot Figure 1B. 

4) The figures have a very low quality.

We apologize when we observed that figures were low quality upon submission. We have high quality figures and hopefully they will appear so in our upload. 

5) Equation 1 – this is not the best way to specify a model. You need to show the dependent variable and the covariates with their coefficients and the error term. Notation is important to follow the paper.

We have modified our expression of our Equation 1 to best address this issue (105-112). 

6) The methods are not clear. They are misleading and they don’t really match what you do in the results. It is important that you rewrite the methods section. The results are interesting, but they need to be well supported by the methods.

We have made large changes and additions to the methods. We hope this clarified how we obtained our results. 

7) I would suggest that you say only once the software used (excel or R).

We have made this change and deleted redundant software references. 

8) While the journal requires you to publish the dataset, I would suggest you create nice tables out of the two excels that you are uploading as supplementary material. They could be very informative. You need to show a descriptive table – number of cases, exported, travelers…etc.

We agree and have added Table 1 with the data sources listed to address this comment.

9) It looks like from the data, that most of the infected cases traveled 4 days before the ban – however, you have a very small sample. Moreover, the incubation period is not known, so these people could have been infected long before traveling.

We agree the sample size from Japan, Singapore, and Korea is not large but we believe it is appropriate and sufficient for sampling purpose (40/~140). The incubation period of these infected travelers is not known and we presume that they were infected before leaving Wuhan. Therefore, we used data weeks after the travel ban to make sure we had the full dataset of exported infected cases. We also mention that Chinazzi’s study using a larger exported dataset had a similar depiction of a critical period of ~5 days in their figure (Lines 185-189).

10) You analyze the exports from Wuhan, but since the outbreak and before the ban, people in China were moving around. Flights from other regions in China to different countries could also carry infected people. Could that be checked too?

We performed analysis on Wuhan travelers because it was a significant contributor to exported cases. Theoretically, we can do the same analysis on other cities if they had a large sample size of exported cases but it does not appear that other cities contributed significantly to the exportation of COVID-19. From the WHO report on February 14th, 2020 [5], there is graph of 200 COVID-19 cases outside China with the suspected method transmission. Exported cases from Hubei province account for over 80% of exported cases from China, so it can be assumed the city of Wuhan accounts for the majority of all exported cases from China. 

11) In the discussion, you talk about a model, but the model is not really explained in the methods section.

We agree and have elaborated on the realistic scenario and model scenario in our methods (lines 119-125).

12) While the discussion is interesting, you need to write down the main limitations. You are looking at the correlation of two variables – so, you might have omitted variable bias. You don’t know the incubation period. And other data limitations that you faced during the study.

We agree and have added many more limitations we faced in our discussions. Some of these include asymptomatic infected travelers, COVID-19 testing methodology, and underreporting (lines 200-203). We do not know the incubation period of the virus in the travelers although we cite literature regarding the COVID-19 incubation period (line 142). However, we do not use the incubation period in our methods except that we account for the fact that there is a delay between the travel ban and the detection of exported cases (lines 245-249). 

13) The paper would benefit from a full rewording – in many cases it feels like the authors are listing concepts (mainly in the methods and results). The text should flow.

We agree and have made significant changes to the methods and results sections. 

Editors Comments

1. Review of the literature: table with other papers examining similar issues. (R1#1, R2#1)

We agree and have added Table 2 to compare our results with other estimates of the Wuhan infected numbers. 

2. Data sources table and output table R2#3 R2#8

We agree and added Table 1 to address this. Additionally, we have made changes to Supplementary Tables to better display our data and sources. 

3. Clarify methods and models. R2#5, R2#6, R2#12

We agree and have made changes to our methods section to address this. We have added clarification to variables and our equation (lines 80-82, 91-94, 99-100, 105-112). We have added an explanation of the models depicted in the supplementary figure (lines 116-122). 

4. Discussion: Please focus your discussion on your results and your limitations. One of them is the incubation period, which according to Bati et al (2020) the median incubation period was estimated to be 5.1 days.

We agree and have added more discussion on our results and of the study limitations. We included the incubation period as a reason our dataset may be more accurate in estimating the infected population in Wuhan at the time of the travel ban compared to previous studies which were published around the time of the travel ban (lines 240-249). We discuss the impact asymptomatic travelers may have on our study (lines 242-245)

5. Future lines of research: Please suggest, based on your findings and limitations, what/how should researchers continue investigating. For example, R2#11 is a relevant issue for further research.

We have suggested in our discussion that government, researchers, and healthcare workers should reassess the infectivity of COVID-19 in an unrestricted environment (lines 253-257). Furthermore, these estimates and research are also useful for future use in case of a new disease outbreak.

---

## [Editor Report · Decision Letter 1]

13 May 2020

PONE-D-20-05233R1

Estimating the prevalence and risk of COVID-19 among international travelers and evacuees of Wuhan through modeling and case reports

PLOS ONE

Dear Mr. Luo,

Thank you for submitting your manuscript to PLOS ONE. After careful consideration, we feel that it has merit but does not fully meet PLOS ONE’s publication criteria as it currently stands. Therefore, we invite you to submit a revised version of the manuscript that addresses the points raised during the review process.

I hope you are well and healthy. Thanks for your revised version, I appreciate that you have addressed all the issues raised during the first review round. I would tend to accept the manuscript, but there are three minor issues that should be addressed before:

1. Add a conclusions section to the manuscript where you summarize your findings and include both policy recommendations and future lines of academic research. My earlier comment #5 has not been addressed fully. You have included policy recommendations in your last paragraph but not what/how should researchers continue investigating. In light of your findings, what should academics pursue or extend from your work? My suggestion was to use R2#11 related to modeling. Your model is a linear regression, which has known limitations (omitted variable bias, heteroscedasticity, unobserved heterogeneity). Please acknowledge this limitation and offer some suggestions to other academics.

2. Provide a roadmap of the paper as the last paragraph of the introduction (that inlcudes the new conclusions section).

I hope you are well and healthy. Thanks for your revised version, I appreciate that you have addressed all the issues raised during the first review round. I would tend to accept the manuscript, but there are three minor issues that should be addressed before:

1. Add a conclusions section to the manuscript where you summarize your findings and include both policy recommendations (you could include the last paragraph in the conclusions) and future lines of academic research. My earlier comment #5 has not been addressed fully. You have included policy recommendations in your last paragraph but not what/how should researchers continue investigating. Respond to this question: In light of your findings, what should academics pursue or extend from your work? My suggestion was to use R2#11 related to modeling. Your model is a linear regression, which has known limitations (omitted variable bias, heteroscedasticity, unobserved heterogeneity). Please acknowledge this limitation and offer some suggestions to other academics.

2. Provide a roadmap of the paper as the last paragraph of the introduction (that inlcudes the new conclusions section).

Since these are two minor issues, I would appreciate if you could send a final revised version before the deadline granted by default.

We would appreciate receiving your revised manuscript by Jun 27 2020 11:59PM. To enhance the reproducibility of your results, we recommend that if applicable you deposit your laboratory protocols in protocols.io, where a protocol can be assigned its own identifier (DOI) such that it can be cited independently in the future. For instructions see: http://journals.plos.org/plosone/s/submission-guidelines#loc-laboratory-protocols

We look forward to receiving your revised manuscript.

Kind regards,

Jordi Paniagua

Academic Editor

PLOS ONE

---

## [Author Response · Author response to Decision Letter 1]

3 Jun 2020

Response to Revision Requests

Editors Comments

1. Add a conclusions section to the manuscript where you summarize your findings and include both policy recommendations and future lines of academic research. My earlier comment #5 has not been addressed fully. You have included policy recommendations in your last paragraph but not what/how should researchers continue investigating. In light of your findings, what should academics pursue or extend from your work? My suggestion was to use R2#11 related to modeling. Your model is a linear regression, which has known limitations (omitted variable bias, heteroscedasticity, unobserved heterogeneity). Please acknowledge this limitation and offer some suggestions to other academics.

We appreciate the suggestion, and we have expanded our manuscript to include a conclusions section with our findings and recommendations (lines 292-304). Additionally, we talk about what researchers and government can do to collaborate for future research (lines 272-291). We acknowledge that our model has limitations (lines 206-213), some of which are explicit (lines 204-206), and that researchers can adapt or develop other models (lines 285-288) depending on data availability. Additionally, we used diagnostic plots to evaluate the assumptions of the linear regression (normality, homoscedasticity, independence, and linearity) which is included in our figures S2 and S3. 

2. Provide a roadmap of the paper as the last paragraph of the introduction (that includes the new conclusions section).

We agree with the comment and have made our last paragraph of the introduction the roadmap of our paper (lines 70-80).

---

## [Editor Report · Decision Letter 2]

8 Jun 2020

Estimating the prevalence and risk of COVID-19 among international travelers and evacuees of Wuhan through modeling and case reports

PONE-D-20-05233R2

Dear Dr. Luo,

We’re pleased to inform you that your manuscript has been judged scientifically suitable for publication and will be formally accepted for publication once it meets all outstanding technical requirements.

Kind regards,

Jordi Paniagua

Academic Editor

PLOS ONE
---

## [Editor Report · Acceptance letter]

11 Jun 2020

PONE-D-20-05233R2 

Estimating the prevalence and risk of COVID-19 among international travelers and evacuees of Wuhan through modeling and case reports 

Dear Dr. Luo:

I'm pleased to inform you that your manuscript has been deemed suitable for publication in PLOS ONE. Congratulations! Your manuscript is now with our production department. 

Kind regards, 

on behalf of

Dr. Jordi Paniagua 

Academic Editor

PLOS ONE